# Eco-Holonic 4.0 Circular Business Model to Conceptualize Sustainable Value Chain towards Digital Transition

**María Jesús Ávila-Gutiérrez \*** , **Alejandro Martín-Gómez** , **Francisco Aguayo-González and Juan Ramón Lama-Ruiz**

Design Engineering Dept, University of Seville, Escuela Politécnica Superior, Virgen de África 7, 41011 Seville, Spain; ammartin@us.es (A.M.-G.); faguayo@us.es (F.A.-G.); jrlama@us.es (J.R.L.-R.)

\* Correspondence: mavila@us.es

**Abstract:** The purpose of this paper is to conceptualize a circular business model based on an Eco-Holonic Architecture, through the integration of circular economy and holonic principles. A conceptual model is developed to manage the complexity of integrating circular economy principles, digital transformation, and tools and frameworks for sustainability into business models. The proposed architecture is multilevel and multiscale in order to achieve the instantiation of the sustainable value chain in any territory. The architecture promotes the incorporation of circular economy and holonic principles into new circular business models. This integrated perspective of business model can support the design and upgrade of the manufacturing companies in their respective industrial sectors. The conceptual model proposed is based on activity theory that considers the interactions between technical and social systems and allows the mitigation of the metabolic rift that exists between natural and social metabolism. This study contributes to the existing literature on circular economy, circular business models and activity theory by considering holonic paradigm concerns, which have not been explored yet. This research also offers a unique holonic architecture of circular business model by considering different levels, relationships, dynamism and contextualization (territory) aspects.

**Keywords:** circular economy; circular business model; holonic systems; activity theory; sustainable manufacturing

---

## 1. Introduction

Circular Economy (CE) or new era of abundance and prosperity is presented as an alternative model to linear economy or scarcity economy [1]. CE is not a new concept; however, the novelty is in the interest of its implementation at the level of government, industry and society. Nowadays, CE is still not contextualized and some authors try to discuss different concepts about this term [2]. The potentiality of CE lies with the purpose of regenerating and restoring the natural environment. Thus, products and processes are redesigned in order to maximize the value of resources. Economic growth is decoupled from the consumption of finite resources [3]. CE is strongly influenced by concepts from different frameworks, such as eco-efficiency [4], Cradle to Cradle (C2C) [5], cyclicity, efficiency toxicity, and Industrial Ecology (IE) [6]. CE is a paradigm of action that has resulted from the evolution of the concept of sustainability [7,8], that includes synergistic articulation from the Triple Bottom Line (TBL), economy, ecology and equity (3E).

The objective of sustainability and CE is to naturify technical systems from the events that occur in the natural world. The main challenge for this approach is to bring the dynamics of natural operation

(from natural ecosystems) to the industrial sector so that the rational use of natural resources takes place according to the rate of renewal, respect and cooperation with nature [9]. From IE, special emphasis has been placed on the pillar of natural sustainability. CE limits the production flow to the level that nature tolerates, using the cycles of natural ecosystems in economic cycles and respecting natural rhythms. This determines that CE conception, as a paradigm for sustainability, needs the research in different pillars such as social sustainability. CE opens new business opportunities (relationships within value chains), innovative challenges and stimulates the emergence of new Business Models (BM). A well-designed BM has a great potential and can serve as an essential strategic tool for the company [10].

BMs are representations of how businesses create economic value for a company through the creation of value for its customers [11]. The incorporation of CE principles [12] to the business gives rise to CBM, where the value is created through the interrelation of activities [13]. Owing to the wide variety of activities and relationships between them, different types of businesses and applications can be found [14–20]. Regarding business, it is necessary to know the relationship between the activities of the system, establishing the means of achieving the objectives (what), the understanding of how to achieve that goal (how), the motivation to achieve the objective (why), and the evolution of every element in the system (where-to) [21]. BMs are used to implement CE and they may have a different relevance according to their scale (micro, meso and macro) [18]. In addition to defining what a BM is, it is necessary to consider dynamic capabilities and interactions that constitute a key issue that enables the creation and implementation of effective BMs [22].

In terms of research, paradigms and frameworks that are trying to address the above issues, are the holonic [23] and the Holonic Manufacturing Systems (HMS) [24], which have had many contributions from research works to doctoral theses, among which are the application of holonic systems to design and manufacture from TBL, creating Eco-Holonic manufacturing systems [25], the holonic concept in automated manufacturing systems [26] or model-driven methodologies for testing a distributed multi-agent manufacturing system [27–29]. These works have allowed to positively evaluate the holonic paradigm to address the variety required by the life cycle engineering of manufacturing systems [30], mainly in different aspects, such as its non-linear dynamic character, its complexity in the different levels of granularity (micro, meso and macro), the variety of views [31–33] (energy flows, materials, water, information, human resources, etc.), as well as the requirements of adaptive innovation to the technological and socio-economic context.

In this work, a highlighted distance from conventional approaches at industrial and social systems is established, as an ecosystem without solution of continuity with natural ecosystems. A particular form of appropriation of material flows, energy and information (social metabolism) provided by the biosphere, from which it cannot be separated, and which would be found in sustainable dynamic equilibrium under a certain endometabolism and exometabolism is considered [34]. The proposed approach considers equally important technology, economic processes, business interrelations, financing, government policies, business administration, society, culture and anthropological aspects based on three elements: (1) a systemic perspective that integrates the components of industry and biosphere, (2) an emphasis on biophysical extraction for human activities and the complex relationships of material and energy flows with industrial system and (3) a consideration of the evolution of the long-term technological dynamics as a transition element to move from an unsustainable industrial system, based on linear economy, towards a cyclical and regenerative industrial ecosystem [1,35].

There are many questions and opportunities in relation to the new era of technical systems [36], among which mitigation or reversal of the metabolic rift [37], caused by the industrial revolution, through emerging technologies and the potential of digitization offered by information technologies in the context of CE and Industry 4.0 are considered [38]. Therefore, this paper addresses the following research questions (RQs): (RQ1) Is it possible to configure technical and social systems (technosphere and sociosphere) of a variety analogous to natural systems (naturesphere) that co-evolve in an adaptive and evolutionarily stable way considering the socio-historical and cultural model from Lev Vygotsky

and Engestrom?; (RQ2) Which is the appropriate paradigmatic framework that allows an open architecture to conceive CBM and their technical systems in a way that are integrated into nature under criteria of sustainability with the TBL from the paradigm of the CE?; (RQ3) Which knowledge, tools, methodologies and techniques should integrate the proposed architecture so that technical system was conceived as nature to obtain the required variety with special emphasis to digital enablers of Industry 4.0?

Regarding the above and the issues and opportunities that arise under the closure of technical cycles, objectives can be formulated based on different issues. Among these objectives are found: (1) The incorporation of sustainability in early phases of economic activity that is formalized through business under the paradigm of CE from TBL, (2) Characterization of CE as a paradigm of sustainability incorporating Activity Theory (AT) as a framework of circular appropriation of natural resources that mitigate the metabolic rift, (3) Reinterpreting the principles of the holonic paradigm from the concept of sustainability, CE, and AT for its projection in the conception of industrial ecosystems, where nature is considered as a model, measure and mentor [39,40], (4) Defining an Eco-Holonic Architecture multilevel and multiscale for business, its value chain, products and eco-compatible manufacturing systems, integrable in the context (territory), (5) Integrating the culture of digitization [41] and connectivity through digital enablers [42,43]. This last consideration can be articulated under the supply chain approach, encompassing every activity associated with the flow and transformation of products from the raw materials (extraction) stage to the end user, as well as the associated information flow [44]. Value chain considers the meaning of value in several contexts, including commercial relationships, consumer purchases and the interests of the company's stakeholders [45]. Lean philosophy is presented as an adequate candidate to define value from the client's perspective.

This approach contributes to give a unified perspective of CBM using tools and frameworks that contemplate the circularity from the TBL. This research is unique if it is compared with studies where businesses are contemplated at different levels but the relationships, dynamics and contextualization (territory) of these are not taken into account [18].

Given the objectives to develop, this paper is organized as follows. Section 2 provides a vision of the BM, frameworks, tools and models for the formalization of business such as AT under the CE and the transition towards TBL in order to establish the necessary requirements that must be satisfied by the holonic paradigm. Section 3, the principles of CE and knowledge base are reinterpreted for their inclusion in the holonic conceptual framework. Section 4, a case study about Eco-Holonic Architecture multilevel and multiscale for a circular business is formulated. Finally, Section 5 presents the conclusions and outlook.

## 2. Background of the Literature

In this section, a review about the main concepts related to this work is carried out. The initial question is related to the existence of CBM that support the different organizations and that constitute an integrating framework of reference from the perspective of the CE.

Based on the types of literature review proposed by Mayer [46], in the present work, a Status Quo review will be carried out. This type of review is being considered as a description of the most current research for a particular topic or field of research that has been structured in the five points shown in Figure 1.

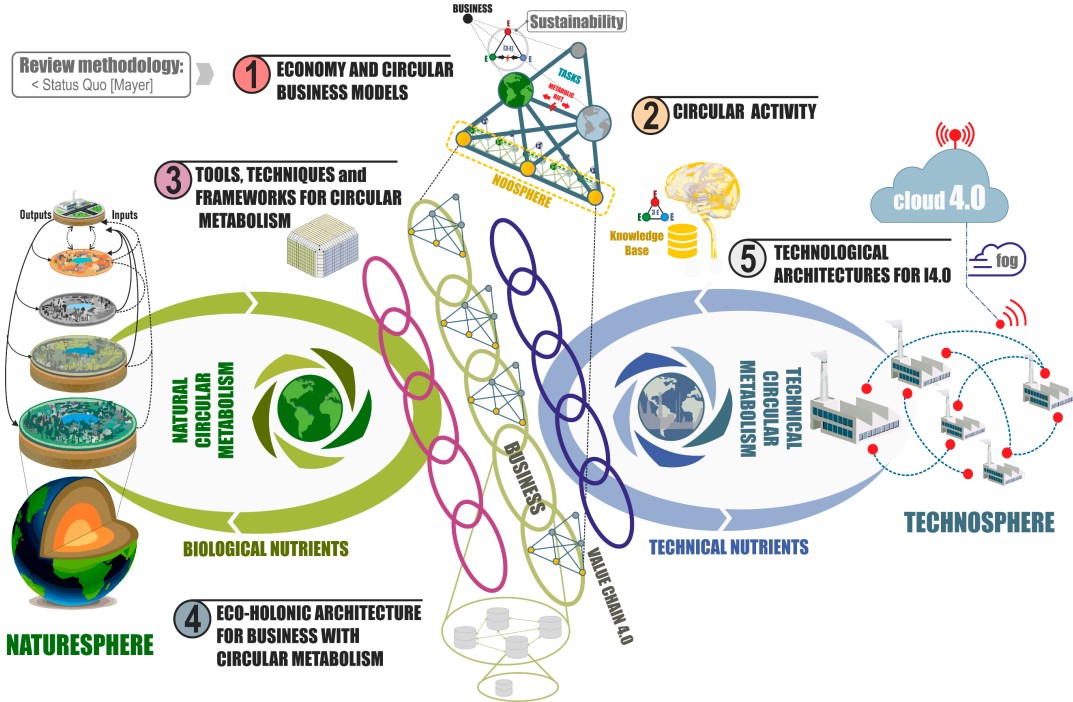

**Figure 1.** Background review.

In this line, several concepts have been analyzed separately as: (1) understanding what is a BM, and which is the last aim pursued by these models; (2) the business as a set of activities that create value; (3) the knowledge incorporated through tools, frameworks and strategies that support BMs from the perspective of CE and 3E; and (4) the selection of the most complete paradigm that satisfies the requirements of current business and its sustainable transition to the CE.

### 2.1. Business Models and Circular Business Model

Companies have to think about how they create and deliver value and which role business models (BM) play. The transition towards new BMs requires to innovate and replace the current BM through the incorporation of new opportunities [47]. There are different approaches about what a BM is exactly, what components constitute it and how it should be used. One of these approaches is presented by Osterwalder and Pigneur [48], who argue that a BM is the reason behind what organizations create, offer and how they capture value. Richardson [49] establishes that the three elements of a BM (value proposition, creation and delivery of value, and value capture) must present circularity, efficiency, effectiveness and safety, in order to achieve optimal performance of sustainability within the CE. Likewise, Geissdoerfer [50] relates how each element of BM is affected by the economic, environmental and social dimensions.

In this way, BMs can hold an intermediate position at a level of abstraction in relation to the form of organization of society for the creation of value [51]. BMs, as a system of interrelated activities, implies that changes in one component often involve changes in others. Thus, making a new product (value proposal) may require new processes and new resources (infrastructure management) [52].

The implementation of the paradigm of CE into the BM supposes notable changes in the same ones [53]; mainly, owing to CE, treats different aspects such as closed material loops through multiple phases [54], CE from IE perspective [55], ecological recirculation and recycling of material resources [56], economic system as a closed material loop in Circular Business Models (CBM) [57], industrial economy of zero disbursements benefiting technical and biological material inputs [58], CE based on commercial models that use economic value retained in products after use [59], economic model in the design and management of processes that seeks to maximize the functioning of ecosystems and human

well-being [60], CE as a core for the reuse of product, component and material [61] and finally, CE as a promoter of high-value material cycles beyond traditional recycling, in order to develop system approaches for the cooperation of producers, consumers and other social actors in the sustainable development. All these possible aspects contribute to promote the change in current BMs, so companies are required to participate in an innovation process of the BM, from a circular perspective [62].

There are many techniques and tools based on frameworks about BM. In this way, some of them are described as the one proposed by Osterwalder and Pigneur, that creates the Canvas BM as a tool to describe, analyze and design current BMs. It describes the logic of how organizations generate benefits, help with structuring the organization and enable the development of strategies in a simple way. This BM is composed of nine blocks that represent the views of the activities of an organization through the processes and value flows carried out in them.

The framework proposed by Al-Debei and Fitzgerald [63] presents four components of the BM: (1) value proposition, (2) value architecture, (3) value network, and (4) financial value. They explain the main constructions and dimensions of the BM concept. The most notable difference with Canvas model is the value network that expresses the cooperative importance of the organization. This model provides a clear vision of the roles of the different stakeholders and how the exchange of value between them takes place.

Joyce and Paquin [64] evolved Canvas model in conjunction with the 3E by adding the social and environmental layer. These three layers are based on 3E, and they are used to measure social, economic and environmental impacts of the activities of organizations, so it has been adopted by organizations with sustainable purposes.

Antikainen and Valkokari [18] found that current BM tools lack some important perspectives on innovation in BMs. Therefore, they add perspectives for CBM such as trends and factors at the ecosystem level, value for partners and stakeholders within a company, and the impacts of sustainability and circularity. This framework suggests a continuous iteration of the evaluation of sustainability and circularity so that the process can be optimized, and its dynamics understood. The framework is the most specified tool for CBM among the existing BM tools. However, it requires further development because the current tool has vague descriptions and limited examples.

Roome and Louche [65] explain how new BMs for sustainability are generated through the relationships between groups and individuals inside and outside of companies. Gauthier and Gilomen [66] analyze the transformations and the nature of change in SBMs. Abdelkafi and Tauscher [67] present dynamic systems applied to SBMs. Likewise, Upward and Jones [68] establish an ontology of SBMs.

Other authors, however, choose Canvas model as a framework focused on the circulation of materials in closed loops, the Business Cycle Canvas [57]. Barquet [69] presents Canvas model as a central component and uses it as a tool for the conceptualization of the circular model. In this way, Dewulf [20] develops an extended Canvas model with the cost and social benefit components.

Among them, the case of Lewandowski [14] is highlighted, which establishes a BM for the CE based on Canvas BM. It identifies how to apply CE principles to the BM framework known as one of the most popular. Although, it should be mentioned that despite the contribution of establishing the principles of the CE to each of the blocks of the Canvas method, it has some shortcomings such as that the model is not contextualized, and it is applied to a static life cycle when the life cycle is affected by external agents so it should be a dynamic life cycle. It shows linear relationships at the same level and not at other levels of specificity, it does not represent the different view of complexity of the business, and it does not represent elements of self-regulation since the business is considered as something in continuous autonomous evolution and self-regulating.

Owing to the great variety of existing BMs, it can be said that each of them shows different development paths, perspectives and interpretations, so that there is no reference model that picks up that variety. Therefore, it is necessary to find a tool that includes the circular values and that contemplates a gap of social aspect of CBM, the instantiation at different levels of concretion and the contextualization in order to achieve Circular and Sustainable Business Models.

## 2.2. Business as Activity that Creates Value

Businesses are systems of activities that create value [47]. CBM is a system of sustainable, intelligent and connected activities that support the principles of CE. Activity Theory (AT) is an approach widely known as the sociohistorical approach to value creation. This theory was the result of the efforts made for the development of a model based on the Marxist philosophy and thoughts. It was initiated by Lev Semiónovich Vygotsky [70]. This research had been expanding by Leontiev and other authors such as Engestrom. Engestrom [71] was a key author in the development of the AT by designing generic templates that have found great industrial application. This theory suggests that the context must be considered in the analysis of human action, and that the object of human action is separated from the result. The key elements in the theory of activity are reflected in Figure 2, and they are: the subject or technosphere that constitutes the technical systems, the object or natural capital, natursphere, and community. Tools, rules and the division of labor constitute the artifacts used in the activities to establish the context. In this case, nature contributes with natural resources to technical systems, which through human action transform natural capital with the use of tools. All of them contextualized in a community, through established rules and a properly organized system. This relationship of natural systems and the society of technical systems has been altered over the years, since in the beginning, their relationship was fully integrated and has gradually been separated by the appearance of work and the artificial. This separation of the natural and the artificial was named metabolic rift, a concept introduced by Marx and continued by Foster [72].

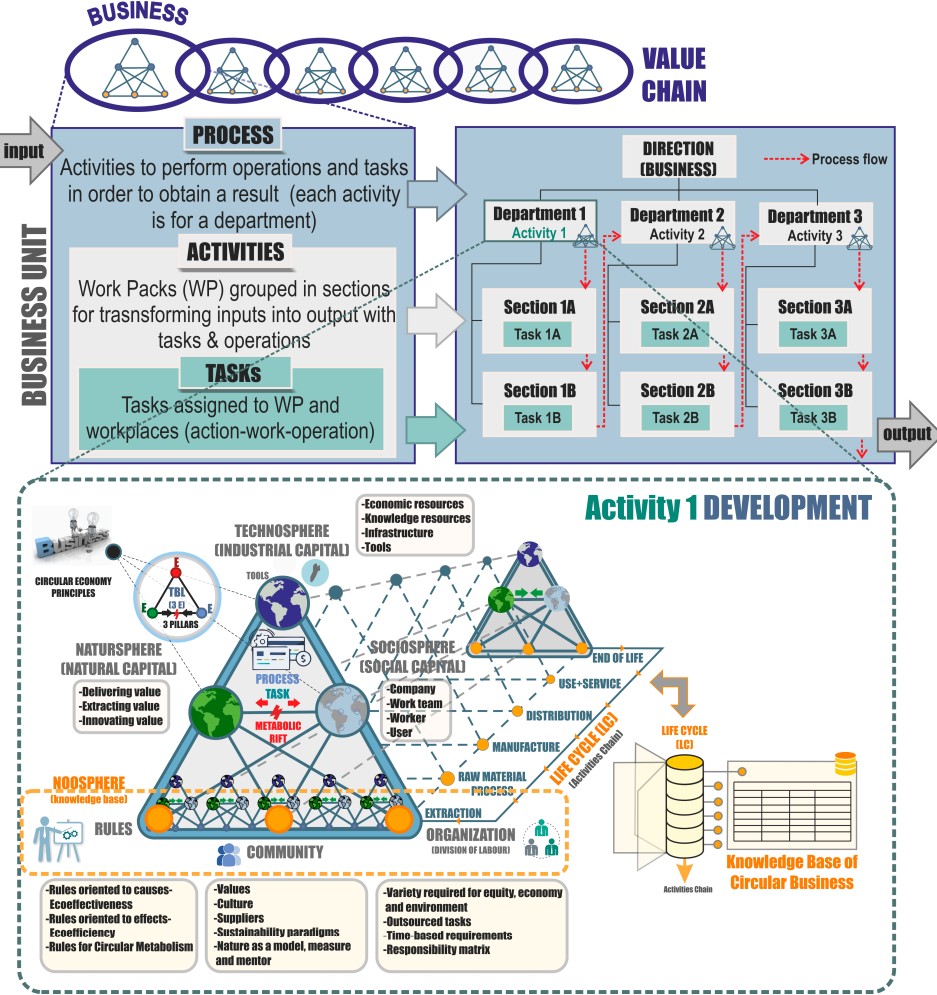

**Figure 2.** Development of business using Activity System.

Research around the AT are substantial and have had many contributions in the fields of education [73] and learning in collaborative environments [74], online learning [75], digital teaching [76], collection and analysis of learning data [77], creativity development methodology [78], information systems [79], apps [80], inclusive education [81], and Building Information Modelling (BIM) implementation [82]. In relation to BMs, there are some works such as designing BMs from different perspectives [83], digital BMs [84], and application to Smart companies [85]. From the point of view of the BM, they do not provide an explicit methodology since they mainly focus on the design and characterization of the different subjects in a qualitative way. They provide a similar approach to a template to ease the analysis of the information gathered from the different nodes as well as the study of possible contradictions.

Actually, BMs are decontextualized. This theory aims to understand human activity as a complex social phenomenon situated where it is composed of the subject, object and tools. All this is developed in a context composed by the community, organization and its rules. From the point of view of a company, it is a generic instrument with a proactive character in the decision-making and evaluation of the processes of an organization, independent of technology and the specific context in which it is carried out. Different decisions can be analyzed and taken to assess the consequences on the pillars of sustainability before the development of the activity.

In addition to AT, as a framework for BM, it is possible to find other frameworks and tools, which are detailed below.

### 2.3. Tools and Methodologies for Circular Economy

CE as a paradigm towards sustainability is based on these causes: lack of cyclicity, efficiency and toxicity in industrial ecosystems. CE uses TBL as the basis for its techniques and tools. CE as a paradigm initially provides a framework that integrates many of the techniques and methods of IE. Life Cycle Assessment (LCA) at its origin was not as a framework for work, but at the present time, CE paradigm integrates different frameworks of work such as value creation frameworks that represent the cause and frameworks that seek to mitigate harm or that represent the consequence [7].

Table 1 shows a list of the tools and methodologies that are used to achieve CE in the CBM. They are classified according to their contribution to the different perspectives, depending on natural metabolism (natursphere) or technical metabolism (technosphere), the aspect of TBL (Ecology-$E_1$, Economy-$E_2$, Equity-$E_3$) and based on its application level (micro (1), meso (2) and macro (3)). The suitability of each tool is represented by ✓ (suitable) and ✗ (not suitable).

**Table 1.** Frameworks, techniques and tools to incorporate the knowledge (noosphere) of AT in CB.

| Methodologies/ echniques | References | Natural Metabolism | Technical Metabolism | TBL | | | Level |
|---|---|---|---|---|---|---|---|
| | | | | $E_1$ | $E_2$ | $E_3$ | |
| Material Flow Analysis | [86,87] | ✗ | ✓ | ✓ | ✗ | ✗ | (1), (2), (3) |
| Life Cycle Assessment | [88–91] | ✗ | ✓ | ✓ | ✗ | ✗ | (1), (2), (3) |
| Material entry per service unit | [92] | ✗ | ✓ | ✓ | ✗ | ✗ | (3) |
| Ecological Rucksack | [93] | ✗ | ✓ | ✓ | ✗ | ✗ | (3) |
| Water circularity | [94] | ✗ | ✓ | ✓ | ✗ | ✗ | (3) |
| Longevity indicator | [95] | ✗ | ✓ | ✓ | ✗ | ✗ | (2), (3) |
| Input-Output Analysis | [96] | ✗ | ✓ | ✓ | ✓ | ✗ | (1), (2), (3) |
| Trophic chain | [97,98] | ✓ | ✗ | ✓ | ✗ | ✗ | (2) |
| Industrial symbiosis | [99,100] | ✓ | ✗ | ✓ | ✗ | ✗ | (2) |
| Ecological Networks Analysis | [101–103] | ✓ | ✗ | ✓ | ✗ | ✗ | (1), (2), (3) |
| Biogeochemical cycles | [60] | ✗ | ✓ | ✓ | ✗ | ✗ | (1) |
| Embedded Energy Analysis | [104,105] | ✗ | ✓ | ✓ | ✗ | ✗ | (1), (2), (3) |
| Emergy Analysis | [106] | ✗ | ✓ | ✓ | ✗ | ✗ | (1), (2) |
| Exergy Analysis | [107] | ✗ | ✓ | ✓ | ✗ | ✗ | (2), (3) |
| Analysis of Social Exergy | [108] | ✗ | ✓ | ✓ | ✗ | ✓ | (2), (3) |
| Sankey diagram | [109] | ✗ | ✓ | ✓ | ✗ | ✗ | (3) |
| Substance Flow Analysis | [110] | ✗ | ✓ | ✓ | ✗ | ✗ | (1), (2), (3) |
| Substance Analysis from C2C | [111] | ✓ | ✗ | ✓ | ✗ | ✗ | (1), (2), (3) |
| Bioinspired or biomimetic design | [112,113] | ✓ | ✗ | ✓ | ✗ | ✗ | (2), (3) |
| Life cycle cost analysis | [114,115] | ✗ | ✓ | ✗ | ✓ | ✗ | (3) |
| Analysis of Economic Networks | [116–118] | ✓ | ✗ | ✗ | ✓ | ✗ | (1), (2), (3) |
| Cost-Benefit Analysis | [119] | ✗ | ✓ | ✗ | ✓ | ✓ | (2), (3) |
| Eco-costs / Value ratio | [120] | ✗ | ✓ | ✓ | ✓ | ✗ | (3) |
| Creating Eco-efficient Value | [121–123] | ✗ | ✓ | ✗ | ✓ | ✗ | (2), (3) |
| Theory of Actor-Network | [124] | ✓ | ✓ | ✗ | ✓ | ✓ | (2), (3) |
| Socioconstructivism | [125] | ✓ | ✗ | ✗ | ✗ | ✓ | (3) |
| Social Life Cycle Analysis | [126,127] | ✗ | ✓ | ✗ | ✗ | ✓ | (3) |
| Social Network Analysis | [128] | ✗ | ✓ | ✗ | ✗ | ✓ | (2), (3) |
| Permaculture | [129] | ✓ | ✗ | ✓ | ✓ | ✓ | (3) |
| Circular Value Ecosystems | [130] | ✓ | ✓ | ✓ | ✓ | ✓ | (1) |
| Natural Capitalism | [131] | ✓ | ✗ | ✓ | ✓ | ✓ | (1), (2) |
| Blue Economy | [132] | ✓ | ✗ | ✓ | ✓ | ✓ | (1), (2), (3) |
| Extended Exergy Analysis | [133] | ✓ | ✗ | ✓ | ✓ | ✓ | (2), (3) |
| Creation of Eco-Efficient Value | [134] | ✓ | ✓ | ✓ | ✓ | ✓ | (2) |
| Industrial Ecology | [135–138] | ✓ | ✓ | ✓ | ✓ | ✓ | (1) |
| Cradle to Cradle (C2C) | [139,140] | ✓ | ✓ | ✓ | ✓ | ✓ | (1), (2), (3) |
| Sustainable Life Cycle Analysis | [141,142] | ✓ | ✓ | ✓ | ✓ | ✓ | (1), (2), (3) |

Derived from this information and in order to organize it, it is possible to establish a toolbox associated to CBM with the components of the TBL, sustainability principles [7], and the view on the natural and technical systems.

*2.4. Eco-Holonic*

The holonic itself constitutes a knowledge framework for the modeling of complex structures in harmony with the context. In the present work, the term Eco-Holonic has been established to reinterpret the interest of the integration of the activity being modeled in a sustainable way. The concept of Eco-Holonical system is the union of the concept of sustainability and holonic system [25]. The central element of Eco-Holonic Architecture is the Eco-Holon. An Eco-Holon is a Holon that is in stable evolutionary equilibrium with the Eco-Holarchies of the Collaboration and Cooperation domains, through a dialectical action. Its bioinspired reference would be an organism that is integrated in a harmonic and stable way in the Collaboration domains. In this domain, transforming the matter, energy and information that constitute it and give it meaning as a living being allow its co-evolution with the environment. An Eco-Holon has properties that are clearly defined in previous publications, such as [143].

In Eco-Holonic Architecture modeling process, the following elements are distinguished: (1) Eco-Holonic entities and Eco-Holarchies; (2) Entities views and generalities; and (3) life cycle model.

1. *Eco-Holonic entities and Eco-Holarchies*. They are the entities that are required to model and project from CE. All entities have a real and virtual part. Depending on the project, it may be necessary to model all the entities or only some of them. Eco-Holarchies are defined in terms of Eco-Holons and

the interactions between them. In Figure 3 is represented an Eco-Holon as a hybrid cyber-physical entity constituted by two parts: (1) a real part with machine to machine (m2m) communication and, (2) a digital twin in cloud with internet connection and great computer capacity. As an example, the real part of the model will be formally specified. A real Eco-Holarchy with h levels is denoted by '$EH$' set defined in Equation (1), where $EH^i$ is the set of all the Eco-Holons at level i of the Eco-Holarchy.

$$EH^i = \left\{EH^i_{Real}, EH^i_{Virtual}\right\} = \left\{EH^0, EH^1, EH^2, \ldots, EH^{h-1}\right\}_{real} \tag{1}$$

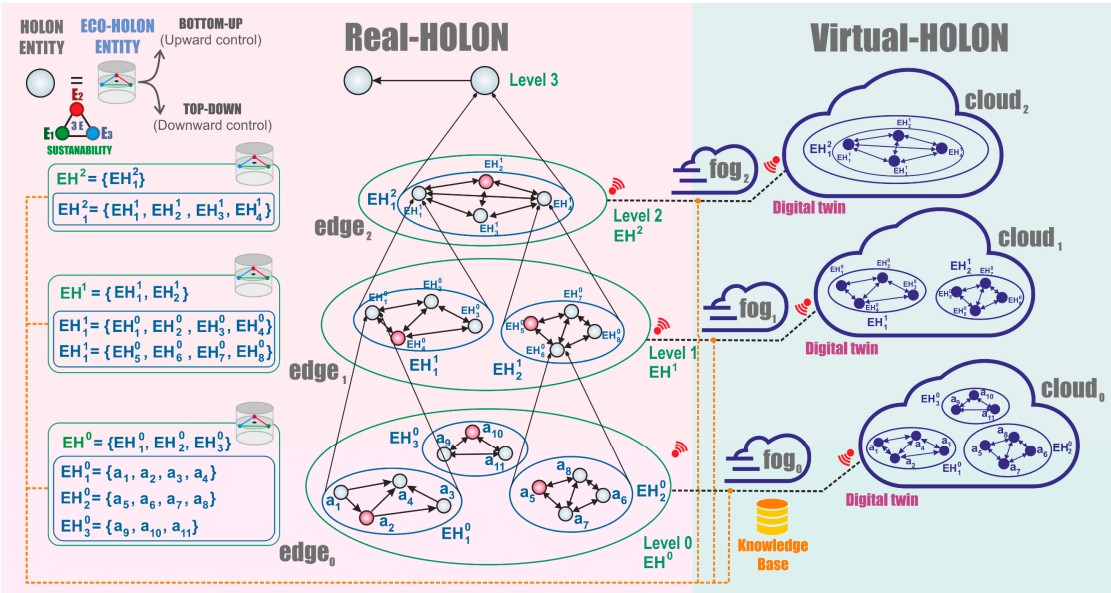

**Figure 3.** Real and virtual Eco-Holon and Eco-Holarchy composed of four levels.

Similarly, a set of Eco-Holons at level i of the Eco-Holarchy is defined by Equation (2), where $EH^i_j$ is composed of Eco-Holon j at level i of the Eco-Holons, and $n_i$ is the total number of Eco-Holons at level i.

$$EH^i_j = \left\{EH^i_1, EH^i_2, EH^i_3, \ldots, EH^i_{n_i}\right\} \tag{2}$$

For this notation, it should be noted that Eco-Holon of level h of the Eco-Holarchy, for example, $EH^h$ is a single Eco-Holon as shown in Equation (3):

$$EH^h = \left\{H^h_1\right\}; \ \forall H^{h-1}_j \in H^{h-1} : H^{h-1}_j \in H^h_1 \tag{3}$$

According to this definition, Eco-Holons at level zero contain Eco-Holons at the atomic level. Formally, these Eco-Holons can be represented according to Equation (4).

$$EH^0 = \{a_1, a_1, a_1, \ldots a_N\} \ o \ \forall a_i \in A : EH^0_i = a_i \tag{4}$$

For this formal specification, we assume that each Eco-Holon at each level of the Eco-Holarchy does not overlap with each other as shown in Equation (5).

$$\forall i, j, k \in \mathbb{N}; i \in [1, h-1]; j, k \in [1, n_i]; \ j \neq k \ : \ EH^i_j \cap EH^i_k = \varnothing \tag{5}$$

For Eco-Holarchy to exist, it is necessary for relationships or interactions between them. Those interactions can be defined as in Equation (6):

$$EHI^i = \left\{EHI^0,\ EHI^1, EHI^2, \ldots, EHI^{h-1}\right\} \tag{6}$$

As shown in Equation (7), for each level i interaction there are:

$$EHI^i = \left\{\left(EH_j^i, EH_k^i\right)\middle|\Lambda_i\left(EH_j^i, EH_k^i\right) = 1; 1 \le j \ne k \le n_i\right\} \tag{7}$$

2.  *Entities' views and generalities.* These views represent the complexity of the life cycle phases of an entity. The views are: (1) Resources view, (2) Informational view, (3) Business process and Operations view, (4) Organizational view, (5) Material, substance and energy flow view and (6) Metabolic view. Regarding to the level of generality, this can be particular, partial (business sector), and general. This approach allows reuse knowledge associated with the grade of view and generality.

3.  *Life Cycle Model.* The life cycle consists of a set of phases which include identification, conceptualization, requirements, preliminary design, detail design, supply of materials, manufacturing, distribution, sale, consumption, withdrawal, disposal, recycling, demolition and elimination. An important aspect is the management of project life cycle, for which are a necessary work environment of design and development such as Product Life Management (PLM) and Building Information Modeling (BIM). Each one of the holonic entities support, in its life cycle, a fractalizable (reproducible in other levels) toolbox with CE strategies and metrics classified in the different phases of the life cycle. This allows to achieve the objectives for CE.

### 2.5. Key Enabling Technologies and Architectures for Industry 4.0

The digital transformation involves digitization and digitalization which affects the whole organization. This determines new opportunities for the business, as well as the search for efficiency with objectives of higher scope in the three dimensions of sustainability, the reduction of the complexity of both static and dynamic manufacturing and the incorporation of intelligent information technology of continuous improvement processes, capitalizing and putting in value the intelligent processes, the skills of the workers, flexibility and restrictions of the technological equipment. This allows us to conceive intelligent factories in closed loop on the Technosphere and is eco-compatible, which mitigates the metabolic rift, creating economic, social and environmental value. To this end, the following will be studied: (1) the appropriate digital enablers for the phases of the value chain and (2) the study of the most significant architectures in the industry 4.0.

### 2.5.1. Key Enabling Technologies for Digital Transformation in Business

In order to monitor and control it in real-time, the concept of Cyber-physical System (CPS) was created. The CPS is the first enabling technology for Industry 4.0, which is configured as an emerging paradigm driven by data and focused on the creation of manufacturing intelligence through the use of ubiquitous networks with real-time data flows. These systems allow objects and processes, which are related in the physical world (robot, numerical control machine, etc.) in the field, to have a virtual representation in the Cloud and in the fog. This allows them to become narrowly coupled and their efficiency to be evaluated through predictive data analysis techniques [144] (e.g., machine learning model) and simulation models from the cyber world in the Cloud as shown in Figure 4.

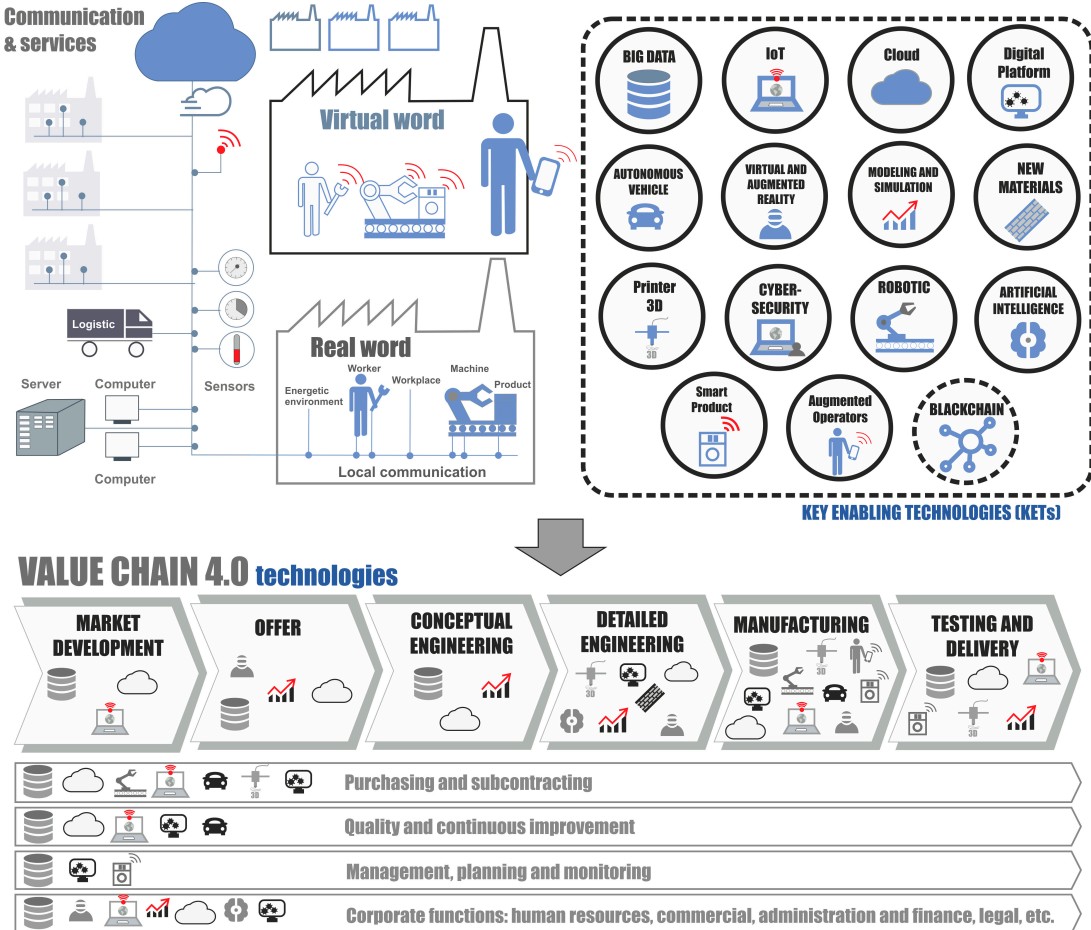

**Figure 4.** Digital enablers and enabling technologies for digital transformation of value chain.

Sustainable business involves the incorporation of the CE paradigm under the principles of TBL and the pillars of sustainability. This proposes the integration in the management and operation of new dimensions, seeking optimization not only in the economic sense as well as in the environmental and social through integrated multiscale and multilevel KPIs.

From the potential of the digital enablers of the Industry 4.0, we have the possibility to conceive the businesses with criteria of sustainability, evolutionary stable and of minimum complexity. Figure 4 shows the most frequently used enabling technologies.

These include the following elements: (1) Big data or massive data acquisition and analysis to predict the behavior of the processes of the value chain and the business; (2) IoT with the aim of providing connectivity and intelligence to different elements of the BM; (3) Cloud or unique storage space for all the files of the program, facilitating access at any time and place; (4) Digital platforms for the connection of different information systems through a common platform, integrating developments from different suppliers; (5) Autonomous vehicles for the optimization of difficult, dangerous and repetitive tasks requiring displacement; (6) Virtual and augmented reality helping operators and engineers when installing, monitoring or designing; (7) Modeling and simulation of processes and systems to predict and analyze their behavior; (8) New materials to reduce weight and increase resistance to adverse effects; (9) 3D printing for in situ manufacture of more complex parts and reduce lead times; (10) Cyber-security or protection against hacking attacks; (11) Robotics or automation of manufacturing, increasing accuracy and decreasing time and cost; (12) Artificial intelligence based on active decision and prediction algorithms, through learning, data analysis and previous results (13) Smart products or products that benefit from novel IT-based approaches to packaging them and their knowledge; (14) Augmented operators or intelligent technology at the frontier of human

development and (15) Blockchain or networks that ensure the certification of transactions, eliminating intermediate agents and added documentation.

### 2.5.2. Reference Architectures for Business in Industry 4.0

Reference architectures for digitized businesses serve as an organizational tool or enabler in relation to their ability to configure intelligent technical and cyber-physical systems. The main architectures for the Industry 4.0 are to be able to establish a vision of what exists and propose a new architecture that improves what currently exists enabling a connected industry of excellent sustainability, co-evolutionary in chaotic and resilient environments. Among the most significant architectures can be mentioned: (1) RAMI 4.0; (2) IIRA; (3) IVI; (4) SME; y (5) BOPI 4.0, and they are showed in Table 2.

**Table 2.** Architectures in Industry 4.0 (I4.0).

| Key | Title [ref.] | Description |
|---|---|---|
| RAMI 4.0 | Reference Architectural Model Industrie 4.0 [145] | This architecture describes the connection between IT, manufacturers/plants and the product life cycle through three dimensions. The perspective of IT projects, which are often very complex and therefore broken down into various components, the product life cycle and the functionalities and responsibilities within a hierarchical organization. Standards-based. |
| IIRA | Industrial Internet Reference Architecture [146] | Reference architecture of Industrial Internet based on standards, which describes the requirements with respect to system, software and enterprise level architecture. |
| IVI | Industrial Value Chain Initiative [147] | IVI offers three perspectives to understand the manufacturing industry as a whole: The flow of knowledge/engineering, the flow of demand/supply and the hierarchical levels from the device level to the company level. Within this architecture, a key element is the introduction of the Smart Manufacturing Units (SMU) which corresponds to one or more general function blocks defined in the model. |
| SME | Smart Manufacturing Ecosystem [148] | This architecture shows a smart manufacturing ecosystem where it includes production, management, design and engineering functions. It shows three dimensions of interest in manufacturing systems as they are (1) PRODUCT, (2) PRODUCTION SYSTEM and (3) BUSINESS. |
| BOIP 4.0. | Basque Open Industry Platform [149] | BOIP is presented as an open interoperability platform for the interconnection of business process data, systems and devices, as well as for the construction of new digital services for industrial organizations. |

From the analysis of the architectures identified for Industry 4.0, no works have been found that provide solutions to aspects such as: organizational enablers that support Lean manufacturing and that can make it evolve to Industry 4.0, the concept of sustainability in any of its views, the management techniques of the variety under minimum complexity, as well as the life cycle in a complete way.

All architecture should have different views, complexity and be able to move on different levels-granularity, geographic scale. The flows or relationships between elements of the architecture must be taken into account. The studies should not remain at some stage of the life cycle but in its completeness. Sustainability is an element to consider since the trend is towards sustainable manufacturing systems. Furthermore, the co-evolution of manufacturing systems and their capacity to recover from external disturbances are also necessary requirements for manufacturing systems in the era of digitalization. Taking into account these premises derived from the knowledge acquired after reviewing general architectures, we establish the comparison of the selected architectures as shown in Table 3, where the suitability of each architecture is represented by ✓ (suitable) and ✗ (not suitable).

**Table 3.** Comparative of Architectures in Industry 4.0 (I4.0).

| | Architectures I4.0 | | | | | |
|---|---|---|---|---|---|---|
| **Requirements** | **RAMI 4.0** | **IIRA** | **IVI** | **SME** | **BOIP4.0** | **ECO-HOLONIC** |
| Views/layers | ✓ | ✓ | ✓ | ✗ | ✗ | ✓ |
| Levels/scales | ✓ | ✗ | ✓ | ✓ | ✓ | ✓ |
| Flows | ✗ | ✗ | ✓ | ✓ | ✗ | ✓ |
| Life cycle | ✓ | ✗ | ✓ | ✓ | ✗ | ✓ |
| Sustainability | ✗ | ✗ | ✗ | ✗ | ✗ | ✓ |
| Adaptability/Regulation | ✗ | ✗ | ✗ | ✗ | ✗ | ✓ |
| Co-evolution | ✗ | ✗ | ✗ | ✗ | ✗ | ✓ |
| Resilience | ✗ | ✗ | ✗ | ✗ | ✗ | ✓ |

From the above, it can be seen that there are many architectures and they are very dispersed. Opportunities are evident in the combined use of some of them to establish a single organizational enabler to support any type of circular business.

## 3. Conceptual Framework for Circular Business

There are important differences between the supply chain and the value chain. It becomes clear that the main difference between both is that the value chain takes the customer as its source and the supply chain takes the product as its origin [150]. In this line, the sustainability of the value chain is an issue that is currently acquiring a growing interest on the part of institutions, companies, universities and consumers [151]. The cooperation between entities that constitute the value chain is necessary to establish the traceability, which will affect quality, reduction of costs, and reduction of safety problems [152]. In order to achieve this cooperation and integration in a sustainable way, it is necessary to establish a framework that includes all the current needs of the value chain. The conceptual framework presented establishes a CBM from the perspective of the holonic paradigm. This model possesses a vertical and transversal integration through the value chain, and it is shown in Figure 5.

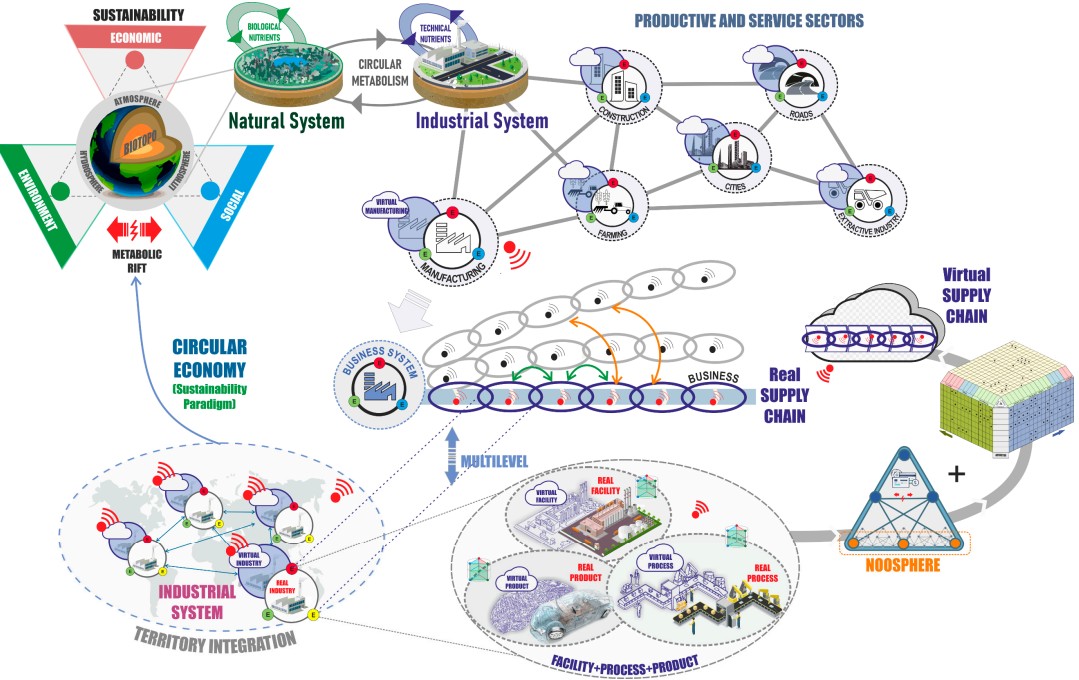

**Figure 5.** An Eco-Holonic Architecture for CB multilevel and multiscale.

The review carried out on the different proposals for the sustainable management of the BM, together with the possibilities of digital enablers of Industry 4.0 [42,43] and Koestler's formulation [23] of the Holon concept (organizational enabler), allow the analysis of complex distributed systems such as the value chain with the incorporation of the potentialities of the digital technologies [153], in order to reverse the metabolic rift [72].

The new CBM must be addressed so that CE would be a success not only for the large multinationals that are at the forefront of innovation, product design and CBM, but also for small- and medium-sized suppliers, recycling companies and other minor entities. In order to achieve this, these new CBMs must be exploited as new ways to transform intensive linear value chains into circular value chains [154].

In order to achieve the transition of the value chain to the sustainable value chain [155], the main needs have been identified and are shown below:

1.  A transition from the value chain to sustainable business 3.0 to 4.0, circular and smart.
2.  An integrated multilevel and multiscale business organization for the required variety.
3.  The incorporation of connectivity and intelligence through digital-digital twin enablers, to develop the intelligence of evolutionary and eco-compatible businesses through innovative horizontal and vertical clusters.
4.  The management of the sustainable value chain from the principles of CE, IE, social ecology, ecological economics frameworks, and the TBL, where economic, ecological and social aspects are integrated and validated through the value chain [156].
5.  The study of the value chain by its geographic and temporal scope from a prospective vision, strategic analysis and open innovation, in the search for collaborative synergistic clusters.
6.  Integration of the value chain in the territory through its contextualization [157]: identifying local synergies, integrating local development, using of local materials, etc. As well as, land use due to the interdependence between infrastructure, production, distribution and environmental resources.
7.  Establishment of vertical synergetic collaborative clusters (between links in the value chain) and transversal ones (between links of different value chains) as driver of open innovation for regional sustainability [158].
8.  Incorporation of a holonic organization enabler [143] for the value chain, conceptualizing collaborative synergistic clusters (vertical and horizontal) as collaboration domains and formulating the cooperation domains of the different links from the holonic principles.

The proposed architecture, based on the holonic paradigm and the principles of CE, allows the contextualization of CBM in a specific territory, where the resilience of the chain is reinforced by the establishment of vertical and transverse integrations from the respective collaborative clusters.

One link of the value chain is represented by a matrix. The matrix contains the point of view of natural (on the left side) and technical (on the right) systems of an activity. The upper part of the matrix shows the synergies between technical and natural systems and their knowledge from CE principles. For the evaluation of this relationship, metrics of sustainability have been selected. Metrics of sustainability bio-inspired are a set of formal and systematic structures and functions that characterize the aspects of robustness, resilience and stability of natural ecosystems for their projection in the evolution of technical systems that ensure and improve the cyclicity, efficiency, and toxicity of technical and biological nutrients. The matrix and its relationship are shown in Figure 6.

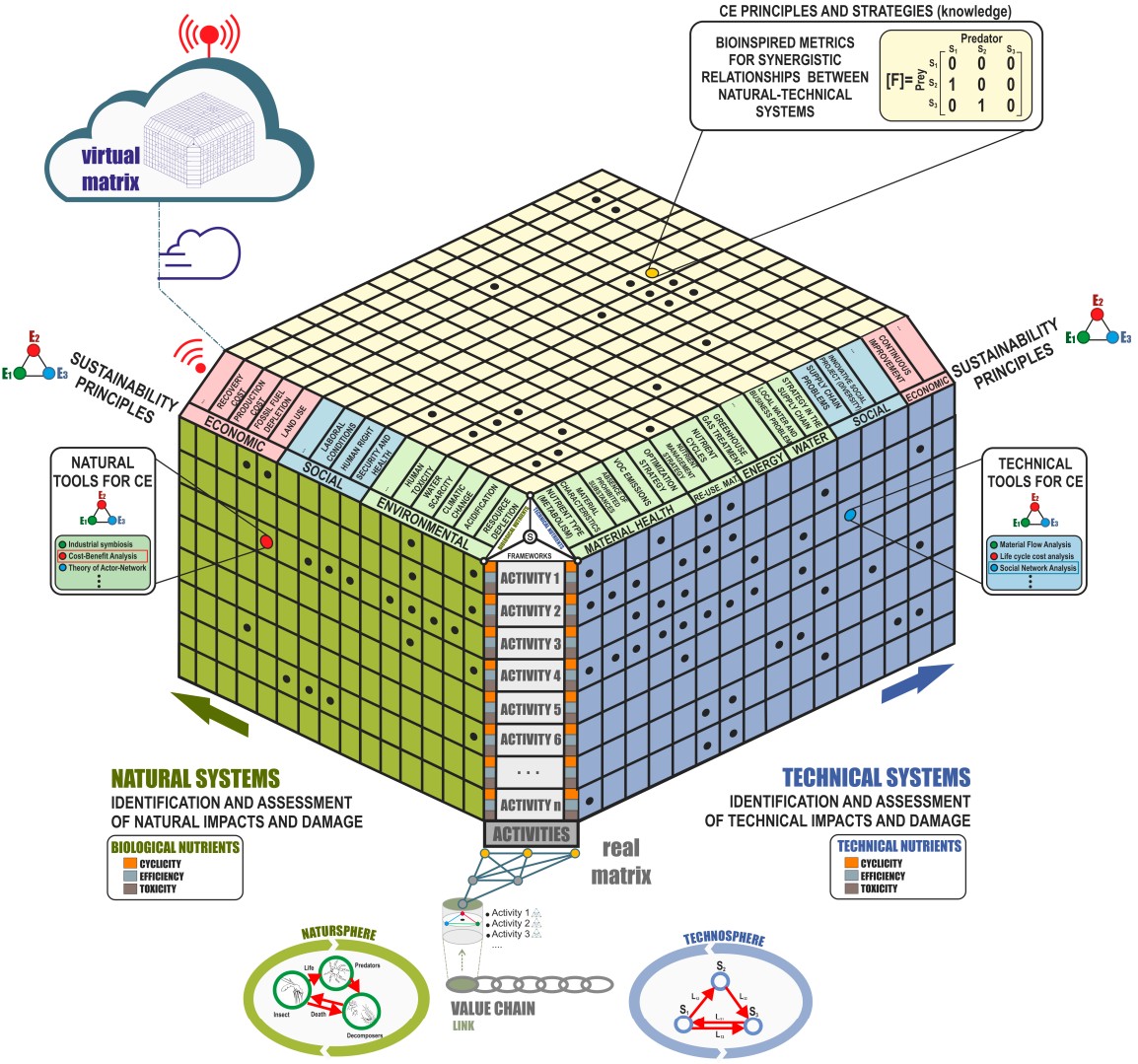

**Figure 6.** Knowledge base matrix for CB.

Among CE Strategies, there is one that is transversal and shares all the stages of the life cycle. This transversal strategy is Lean manufacturing [159]. This is oriented to sustainability and its aim is to optimize the Eco-Holon in the use of material, water and energy resources. In order to carry out each one of the strategies incorporated into Holon knowledge (noosphere), it is necessary to have tools that allow to carry out the strategy for each specific process of the life cycle, with the characterization of the industrial metabolism for CE.

The proposed holonic structure is a fractal configuration that parameterizes its elements under the situation and context of application, to form ecosystems in which the Technosphere (technics systems) is integrated into the Biosphere (natural systems).

AT, tool matrix and Eco-Holon life cycle, contain the knowledge of the Holon, as shown in Figure 7.

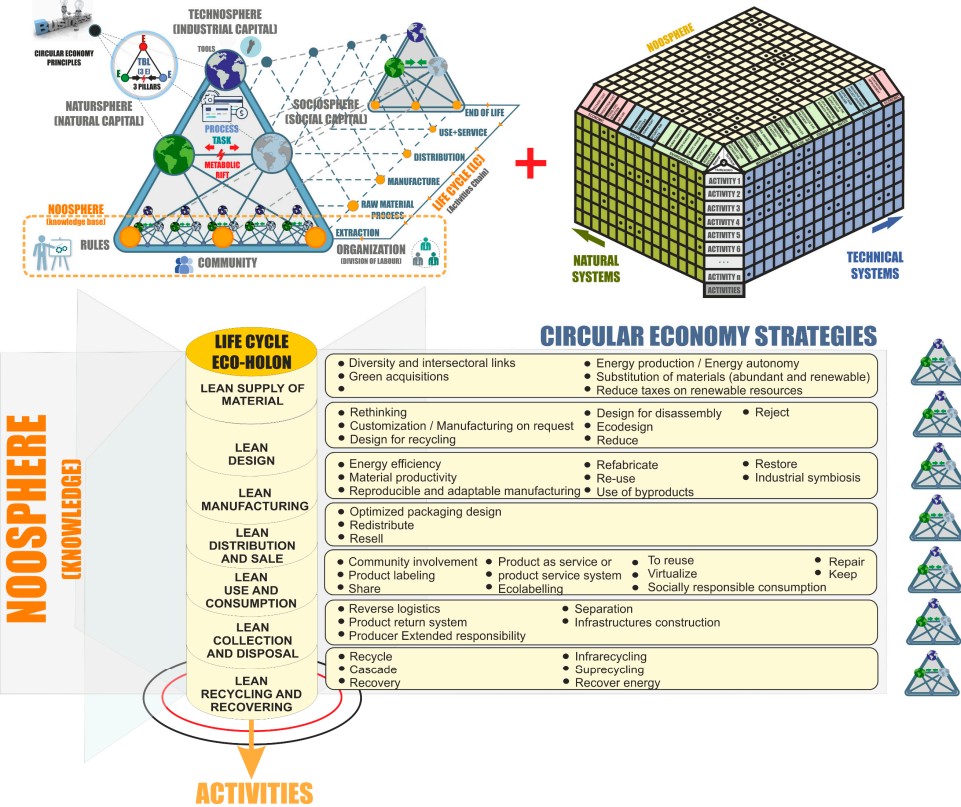

**Figure 7.** Knowledge base in the Life cycle Eco-Holon for Circular business.

The value chain works as an interface between natural and technical systems. Each link of the value chain can be conceptualized as a matrix and to evaluating the relationships between links are used the metrics.

Many metrics have been developed to understand the relationship between the structure and behavior of each link of the value chain from the point of view of naturifying the systems [160,161] as shown in Figure 8 (industrial ecosystem adapted from [98]).

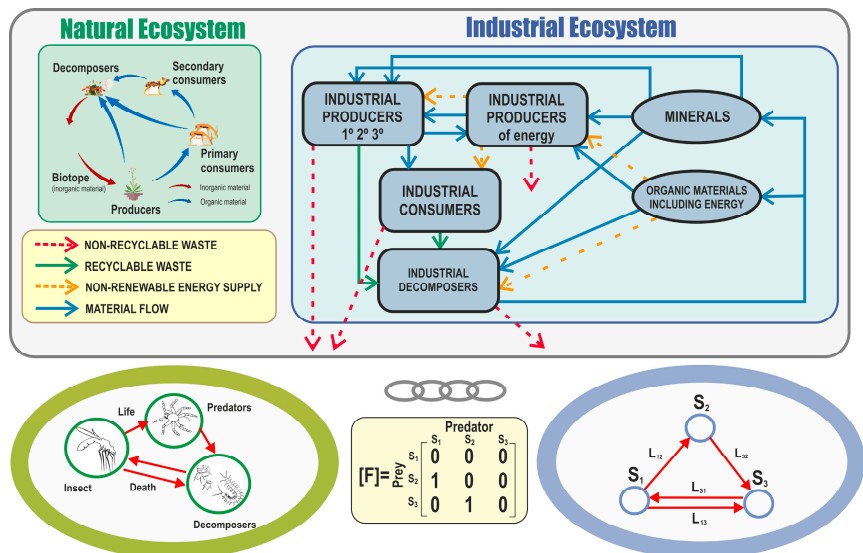

**Figure 8.** Ecosystem, relationship and matrix representation.

The structural measures and metrics most often used by environmentalists can be calculated using binary matrices of the structures. All calculations are based on if a relationship exists or not, taking 'zeros' and 'ones' as data and completing the relationships between the actors that are related.

The following metrics are commonly used by ecologists to conduct Food Chain or Food Web assessments [162]:

- Number of species or actors (N). The total number of actors in a network, sometimes referred to as 'species richness'. It is represented by the number of rows and columns in the trophic chain matrix [F].

$$f_{row}(i) = \left\{ \begin{array}{l} 1 \ for \ \sum_{j=1}^{n} f_{ij} > 0 \\ 0 \ for \ \sum_{j=1}^{n} f_{ij} = 0 \end{array} \right\} \tag{8}$$

$$f_{column}(j) = \left\{ \begin{array}{l} 1 \ for \ \sum_{j=1}^{m} f_{ij} > 0 \\ 0 \ for \ \sum_{j=1}^{m} f_{ij} = 0 \end{array} \right\} \tag{9}$$

- Number of links (L). The number of direct links or interactions between actors in a network. Represented by the total number of non-zero interactions in the matrix [F].
- Link density ($L_D$). The ratio of the total number of links to the total number of agents in a network.

$$L_D = \frac{L}{N} \tag{10}$$

- Number of prey ($n_{prey}$). Actors who are eaten by at least one other. Represented by the number of non-zero rows in the matrix [F]. 'Prey' in an industrial network transfers material or energy to be used by others; i.e., they are the producers.

$$n_{prey} = \sum_{i=1}^{m} f_{row}(i) \tag{11}$$

- Number of predators ($n_{predators}$). Actors that eat at least one other. Represented by the number of non-zero columns in the matrix [F]. 'Predators' in an industrial network receive matter or energy from others; i.e., they are the consumers.

$$n_{predator} = \sum_{j=1}^{n} f_{column}(j) \tag{12}$$

- Prey-predator relationship ($P_R$). The relationship of the number of actors eaten by another with the number of actors that feed on another. That is, the proportion of prey-predators or producers-consumers. This is the number of non-zero rows in the matrix [F] between the number of non-zero columns.

$$P_R = \frac{n_{prey}}{n_{predator}} \tag{13}$$

- Specialized Predator Fraction (Ps). The number of predators (consumers) eating only one actor, divided by the total number of consumers in the network. This is the sum of the number of columns with only one non-zero element in the matrix [F] divided by the total number of columns with non-zero elements.

$$f_{s-column}(j) = \left\{ \begin{array}{l} 1 \ for \ \sum_{j=1}^{m} f_{ij} = 1 \\ 0 \ for \ \sum_{j=1}^{m} f_{ij} \neq 1 \end{array} \right\} \tag{14}$$

$$n_{S-predator} = \sum_{j=1}^{n} f_{s-column}(j) \tag{15}$$

$$P_S = \frac{n_{S-predator}}{n_{predator}} \tag{16}$$

- Generalization (G). The average number of preys eaten by predators in a network, which corresponds to the average number of producers interacting with a consumer in an industrial network. Generated by adding the columns in a matrix [F] and dividing this figure by the number of columns with non-nil elements (the number of predators/consumers).

$$G = \frac{L}{n_{predator}} \tag{17}$$

- Vulnerability (V). The average number of predators per prey in a network, which corresponds to the average number of consumers interacting with a producer in an industrial network. Generated by adding the rows in a matrix [F] and dividing by the total number of rows with non-zero elements (the number of prey/producers).

$$V = \frac{L}{n_{predator}} \tag{18}$$

- Cyclicality ($\Kappa_{máx}$). It represents a measure of the force and presence of cyclic pathways in the system. It is obtained by finding the maximum actual eigenvalue of the transposed matrix [F].

$$C = \frac{L}{N^2} \tag{19}$$

$$\Kappa_{máx} = maximum\ real\ self - value\ solution\ of\ 0 = \det(A - \Kappa I) \tag{20}$$

The formulas for these parameters are represented by the Equations (8)–(20) representing the relationship between the actor's rows and columns. All calculations represented by the equations are based on binary information of whether or not there is a link between two actors in the matrix, which makes these metrics very suitable for application to manufacturing networks, as it eliminates the need to obtain material and energy data often owned internally by manufacturers.

## 4. Study Case: Eco-Holonic Architecture 4.0 for Circular Business

This section is divided in two parts. Firstly, a generic circular business is contextualized under Eco-Holonic framework and secondly, a particular modeling framework for the informational view of Eco-Holonic Architecture is carried out.

### 4.1. Conceptual Eco-Holonic Architecture for Circular Business 4.0

The conceptual framework establishes an Eco-Holonic Architecture (multiscale and multilevel) according to CE principles. This architecture is shown in Figure 9 and it is integrated by a principal Holon called 'Circular Business Eco-Holon' and six Eco-Holarchies: (1) circular sector; (2) circular value chain; (3) circular industrial plant; (4) circular resources; (5) circular manufacturing, and (6) circular device.

The principal Holon is Circular Business Eco-Holon. It is an abstract Holon that represents the business in itself and that encompasses everything related to the business at any level and scale as the activity carried out to obtain a benefit. Circular Business Eco-Holon is whole of Circular Sector Eco-Holarchy, that is composed for the Eco-Holons of agricultural, industrial, construction and smart city sectors. These sectors are not unique and other sectors could be incorporated. Firstly, the agricultural and industrial sector, represent the first stage of metabolism, linked to the appropriation of environmental resources. Secondly, the construction sector, oriented mainly to satisfy the exosomatic metabolism of society's metabolism, referred to the structures used by individuals to satisfy their needs

and interests. Finally, the smart city is identified as a preferred consumer of goods and services. This Eco-Holon, as an abstract entity, collects the principles and objectives of CE and natural principles on which it is based. Other holons in the successive holarchies maintain these principles consigned in this Holon.

The holarchy must be self-organized and in real-time to mitigate and reverse the metabolic rift. According to this objective, the concept of circularity of CE [163], acquires importance in the definition of a strategy that must be carried out by the holarchy. The circularity of resources is necessary for improving the economic and environmental performance of industrialized countries [164]. This is related to the process of circularity in the industrial metabolism model through the value chain.

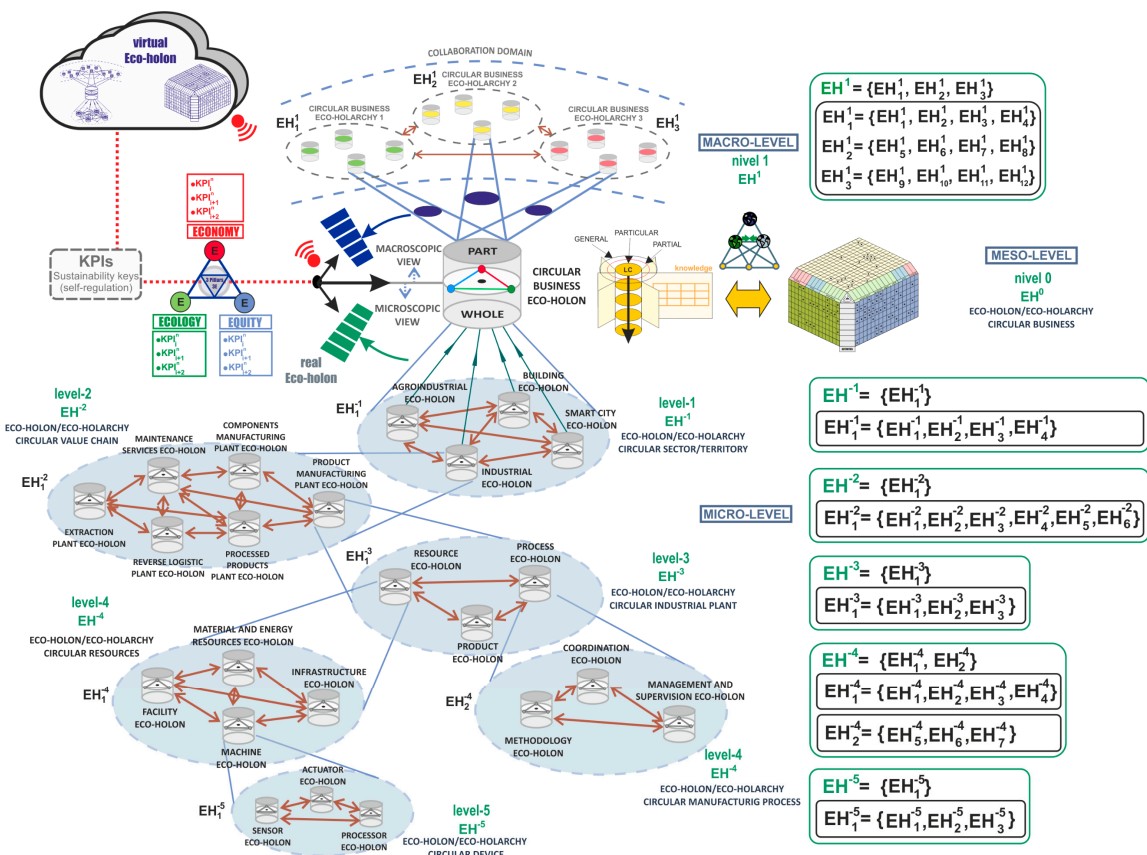

**Figure 9.** Eco-Holonic modeling entities for Circular Economy Business as Cyber-physical entity.

In Figure 9, this holarchy aims to integrate the links of the value chain in Cooperation domain, to promote the coevolution of the value chain through vertical and horizontal integration. The Eco-Holarchy of Circular Value Chain contains associated techniques as Ecological Network Analysis (ENA). Depending on the flow of resources (substances, materials, energy, etc.), ENA is used to study the ecological relationships between holons [165]. The characteristics of the metabolic network that constitutes the holarchy and the natural, social and economic components of an industrial ecosystem can be abstracted as nodes in a network, and the exchanges between these nodes can be treated as metabolic pathways [102]. The point of view of the analogy between natural and social metabolism and the relationship between two holons can be of different types [166]. Consequently, the knowledge of ENA is incorporated into the holarchy, which uses ENA to drive the flow and analyze the metabolic pathways. Therefore, the analysis tools can be used to analyze the ecological relationships between holons [102,167,168] in a holarchy or in several holarchies at the same level.

The Circular Engineering Company is based on models of business architectures used as instruments that allow companies to face the associated challenges with the complexity of the

operating environment. Among business architectures, the most known are [169]: CIMOSA, GRAI, PERA, ARIS, MISSIOM-IEM, AKM and ArchiMAte.

The CBM has its origin in the engineering company, which is developed through a methodology and design and development work environments as BIM and PLM.

A fundamental component of Circular Engineering Company Eco-Holon is the Methodology Eco-Holon. This Eco-Holon takes part in a fractalized way at different levels (creation of a company, projection of the industrial park, industrial plant and product), giving support for the design of processes, strategy and its management, as well as the integration with the technological and information systems.

CBM, in relation to the industrial plant, must be instantiated in the territory, under criteria of diversity and principles of cohesion and organic development. For this, it is necessary that there is a CBM entity that manages the use of land and the common infrastructures that can supply various business established in a specific geographical area. This holon can also be called, depending on its embodiment in a specific territory, Industrial Park Eco-Holon. The fundamental objectives of these Eco-Holons are to facilitate materials, substance and energy closing cycles of products-processes, and to maximize the three value components of value chain that forms the cluster of CBs. All this through the decision to locate the industrial plant using optimization tools from the perspective of industrial metabolism.

Circular Industrial Plant Eco-Holon manages the internal processes to the industrial plant and the relations with the rest of holons of industrial plant. The internal holarchy has a fractal character as a product, process and resource. It is composed of product entities, strategies and procedures to encapsulate the complexity of processes, equipment, facilities and infrastructure resources. These three entities make up each link in Circular Value Chain.

Circular Product Eco-Holon has knowledge of the product and its manufacturing process to ensure the correct manufacture. Additionally, it is based on the principles of CE. It contains information about the product life cycle, user requirements, design, process plans, material list, quality assurance procedures, and models of the type of product. It acts as an information server for the other Eco-Holons in the holarchy. This holon comprises functionalities that are traditionally covered by product design, process planning and quality assurance. Likewise, it assumes the objectives of evaluation and monitoring of the product in its life cycle and includes the three dimensions of sustainability, the reduction of entropy based on the components, processes incorporated in its life cycle, and the orientation towards the closure of cycle. This Eco-Holon contains information on the type of nutrients (technical or biological) that are incorporated, useful in the establishment of metabolic pathways.

Circular Process Eco-Holon has the knowledge of the production process to ensure the correct manufacture of the product based on the principles of CE. It contains information and methods to manufacture the products using resources. Circular Process Eco-Holon monitors evaluates and transfers information to other holons of the holarchy. This information is about KPIs in the environmental dimension (cyclicity, efficiency and toxicity of process), social and economic. This information allows the rest of Eco-Holons of the Eco-Holarchy to make decisions based on their objectives under the principles of CE.

Circular Resource Eco-Holon, as a cyber-physical system, contains a physical part, a production resource of the manufacturing system, and a part of information processing that controls the resource. It offers production capacity and functionality for Eco-Holons of Eco-Holarchy. It contains methods for allocating production resources and knowledge and procedures to organize, use and control these production resources. A Circular Resource Eco-Holon is an abstraction of the means of production, such as a factory, a workshop, machines, conveyors, pipes, self-guided vehicles, components, tools, material storage and personnel.

### 4.2. Technological Implementation of the Eco-Holarchies in Circular Business

The modeling framework of the informational views of the different holarchies of the cyber-physical holons will be done under the modular Arrowhead architecture which has an orientation to

microservices. Arrowhead enables different modes of client-server relationship, allowing the integration of new components (sensors/actuators) and facilitating interoperability with other systems. It is based on the use of containers which allow a deployment of services from a base container image. The structure of a basic holon from the point of view of microservices is as shown in the Figure 10.

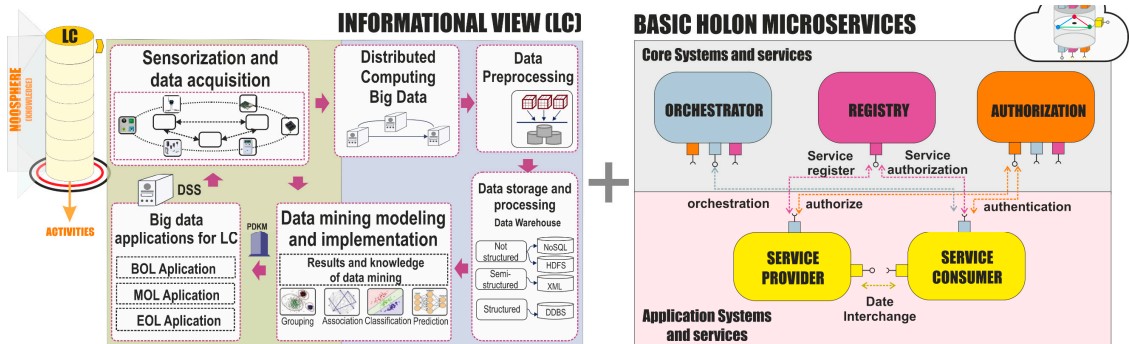

**Figure 10.** Informational view and basic Holon microservices.

This contributes to the isolation of services from the architecture, separating it from other services and providing higher levels of security. The owners of these services are small, independent teams that can be supported in the Cloud. The arrowhead architecture together with the docker technology, in addition to the features such as: Agility, scalability, fractality, modularity, autonomy, specialization, simplicity, reusability, and security, makes it the right approach to implement the information view of the Eco-Holonic Architecture of the circular business.

Figure 11 shows a multilevel and multiscale Eco-Holonic Architecture for the circular business with three views: (1) conceptual view, (2) knowledge view, and (3) information view. It constitutes the three specification and modeling domains corresponding to the Domain of the Circular Eco-Holonic Business, the Domain of the Eco-compatible metabolic knowledge on the technosphere and the naturesphere, and the Domain of the information system for the optimization and continuous improvement of the metabolism from the excellence of sustainability, respectively.

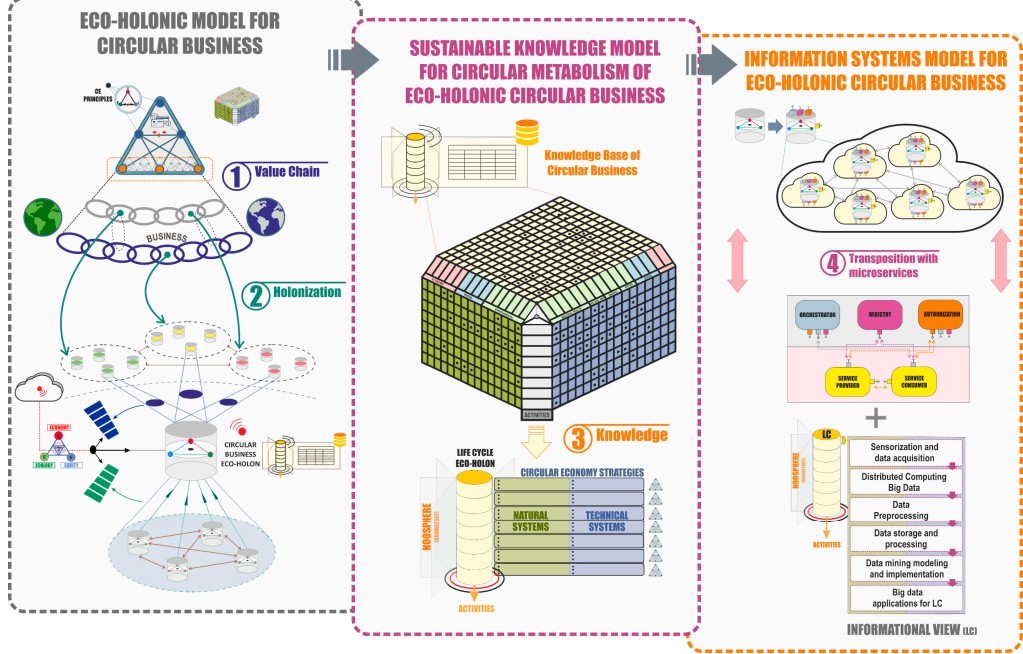

**Figure 11.** Technology implementation for Eco-Holonic model.

The most significant features of these domains are: (1) Establishment of the design domain holarchies in the context of its sustainable value chain from the paradigm of the CE throw study of value chain an holonization of the business entity; (2) establishment of the knowledge model that will support the holon at any of the levels and scales for the development of sustainable business activity under the paradigm of the CE and (3) formulation of an information system based on microservices for the Eco-Holonic Architecture of Circular Businesses 4.0 throughout its life cycle.

4.2.1. Domain of Holarchies' Design in the Context of the Sustainable Value Chain from CE

This domain is articulated under the consideration of strategies on the collaborative domain and the cooperative domain and the self-regulatory mechanisms of the holons [143]. To this end, it is necessary for:

1.  Identification of the characteristic features of the context of the circular business that constitute the domains of collaboration or accommodation holarchies and formulation of the natural and social territorial axes in which these holarchies will be integrated within the business. This implies a wide knowledge of the reception ecosystems and the establishment of an environmental, social and economic ecological inventory that will be affected by the business to be implemented.
2.  Identification and characterization of the collaborative domain formed by the potential longitudinal collaborative clusters of the value chain in which the business is integrated, under the consideration of cyclicity, toxicity, efficiency on the technosphere and naturesphere ecosystems in the search for eco-compatibility and regenerative metabolism in the business with biological and technical nutrient analysis.
3.  Identification of transversal collaborative clusters with other holons (links) of the holarchy of the value chains for open innovation, having as objectives eco-compatibility and regenerative metabolism from the analysis of the principles of the theory of sustainable activity under CE paradigm.
4.  Integrated multilevel and multiscale development of business holarchies and their cooperation domains to guarantee self-regulation from the eco-metabolic CE principles under the three pillars of sustainability through KPIs, structured in the three dimensions of 3E, in an integrated way and oriented to the continuous improvement from the excellence of sustainability constituting the consignments of the holonic entities.
5.  Conception of the features of the Eco-Holonic entity of the business as cyber physical systems in its double real and virtual world existence with regard to skills and knowledge for the improvement, optimization, monitoring and control of sustainability, establishing knowledge, KPIs, improvement strategies, balanced scorecards and subordinated models, in the cloud, in the fog and in the field.

4.2.2. Establishment of the Multilevel and Multiscale Holon Knowledge Model

Once formulated, the multilevel and multiscale holarchies of the Eco-Holonic conceptual domain of the circular business, it becomes necessary to establish the knowledge model that will support the holon at any of the levels and scales for the development of the business activity, according to the level of operational concretion proposed from the theory of the activity, which offers a great opportunity to conceive the sustainability of the business, with its processes and tasks. The objective of this stage is the conception of Eco-Holonics businesses with an eco-compatible metabolism with the naturesphere and technosphere in addition to remediate the metabolic rift. For this purpose it is necessary to:

1.  Determine the activities to be carried out in the different stages of the holon's life cycle, defining the different elements that characterize it for the purposes of cyclicity, toxicity and efficiency in the ecosystems of the technosphere and the naturesphere.
2.  For the different stages and activities to identify the methods, techniques and tools to obtain the KPIs of the three pillars of sustainability under CE paradigm, formulating an instance of

the knowledge matrix for the eco-compatible metabolism of the business with the naturesphere and technosphere.

3. Determine a set of applications and software tools in the different stages of the life cycle for the eco-compatible metabolism of the business in a modular way to be supported by the architecture of the service-oriented information system.

4.2.3. Formulation of An Information System for the Eco-Holonic Architecture of Circular Businesses 4.0

The establishment of an information system for the Eco-Holonic Architecture of Circular Businesses 4.0 must allow the development of the Sustainable Life Cycle Engineering from the paradigm of the CE, oriented to the excellence under strategies of monitoring and control through the control panels of the KPIs in its three pillars of sustainability. For this purpose, this information system requires:

1. In the design and development phase of the business, it is necessary to have methods and tools for the modeling and simulation of the business, formulating digital models and simulations of Products, Manufacturing Processes in PLM and Facility environments in BIM applications, which will later constitute the digital models that will be integrated into the digital twin. Together with the business holon model and the product, process and facility holons, it is necessary to conceive the associated information systems as a cyber-physical Eco-Holon Architecture in the dimensions of the Cloud, Fog and field.

2. In the third part of Figure 10, the information system associated with the Eco-Holonic Architecture of Sustainable Business from the CE is represented [170]:

   a. Sensorize the cyber-physical entities in their real dimension in the field through intelligent sensorics, digitizing the sustainable behavior at the operational level.
   b. Data entry, cleaning and processing and storage in SQL and NSQL databases.
   c. The treatment with techniques of Artificial Intelligence, Big Data, Data Anality, Data science, Cognitive Computing, machine Learning, deep learning . . . formulation of subrogated models for obtaining and controlling sustainability, visualization of sustainability KPIs in different control panels of the business holarchy.
   d. Formulate knowledge models for sustainable business management and operation.

Everything mentioned is based on operational, development and process concepts, as well as on the organization in distributed environments and using Cloud services. Microservices are an architectural and organizational approach to software development where the software is composed of small independent services that communicate through well-defined APIs. Microservices have a greater practical orientation, due to having lighter mechanisms than SOAs, that is, in a more efficient way. Nevertheless, there is an equivalence with service-oriented architectural styles, where microservices are the components, and the connectors represent the interactions between them.

Another important topic is coordination mechanisms. From the point of view of microservices, coordination can be done in two ways: by choreography or orchestration. Choreography prevails over orchestration commonly used in traditional service-oriented architecture such as SOA. In choreography, each node listens to events (event-driven), and performs its part if necessary, by originating 'Service Call Chains', while, in orchestration, there is a central node that controls the services that are called. For interaction between services by choreography, each microservice is required to know details of the domain including the endpoints, and the communication protocol.

In terms of the deployment/operation, and because of the importance of the execution environment in technology, operational aspects, including information on the instances of packaged services, and their behavior must be considered. The characteristics or attributes that should be associated with dockers that run a microservice are mentioned in the study [171]: IP, Identifier, Network, Interface.

Dockers' use in the definition of a microservice architecture is important as expressed in several researches [171–174]. Although a docker is similar to a virtual machine, it shares the same operating system as the machine where the docker is running. Although it is dependent on the host machine, it improves performance and speeds up system startup.

An essential requirement among local clouds is the ability to communicate safely. To this end, the Arrowhead framework provides local numbers for the monitoring and control of a given machine or local clouds associated with the management of a process or even integrated into a service-based business management architecture, easily solving the inherent interoperability problems.

In practice, the successful implementation of the proposed framework would require the necessary previous requirements to implement the Eco-Holonic model. Among the necessary conditions are found businesses that determine: (1) Small, medium or large companies in the process of implementation or digital transformation and integration of KETs; (2) The adoption of an open innovation culture model for the establishment of vertical and horizontal clusters in the virtual value chain; and (3) The conception of a collaborative, self-assertive and intelligent information system architecture based on cyber-physical systems that have as strategic objectives the incorporation of sustainability under the paradigm of the circular economy.

## 5. Conclusions

This work provides an Eco-Holonic CBM to contextualize the sustainable transition towards CE. The proposal aims at the incardination within paradigm of CE, based on its pillars (economy, ecology and equity) in a way that allows mitigating or reverting the metabolic rift. All this from the possibilities of eco-efficiency (effects or reduced damage) and eco-effectiveness (causes or added value). In this way, by the synergistic articulation of eco-effectiveness and eco-efficiency, it is possible to design the social metabolism of CB and their value chain or supply.

A novel framework is presented for the analysis of the elements or activities that incorporate value in the CB based on the AT of Vygotsky's, on which has settled in its noosphere (knowledge) the set of techniques and tools which are synergistically articulated to solve the contradictions of circularity between the activity elements with the aim of mitigating the metabolic rift. AT is integrated into the value chain of CBM, with a proposal of transition of value chains 3.0 to 4.0. This framework works as an organizational enabler that supports the digitalization of the value chain of a business as a cyber physical system (Industry 4.0).

Eco-Holonic Architecture is proposed for the development of CB and their technical systems, developing the ontology of the different holons and holarchies. In this context, the holonic ontology of CB is proposed and allows the business deployment in an industrial territory (contextualization). The proposal is an innovative contribution to the development of CBM in the context of the emerging paradigm of CE, digitalization of businesses and companies, and the aim of mitigating the metabolic rift from the early stages in an aggregate level as it is the business value chain. The quantification of this model could be incorporated into the holarchies establishing the relations and flows between entities with quantitative methods such as ecological relations [98,175], living system theory in the different levels of granularity for the modeling of the knowledge [176–178], contextualizing material, energy and information of the process and use of sustainable indicators for each of the entities [179]. Another aspect to consider is the incorporation of open innovation with fractal criteria (repeated at different scales) in Eco-Holonic Architecture, and agile and Lean principles in the business [180,181]. Moreover, this paper proposes the Arrowhead architecture as a tool for the creation of local clouds to enable real-time performance and security, interoperability and scalability through cloud interaction. To this end, the proposed Eco-Holonic conceptual framework has been transposed into the domain and functional information modeling with the elements of the Arrowhead architecture and subsequent implementation with Docker technology. This favors a new efficient, effective and smart business in terms of reducing inputs of natural resources, material and energy, or waste and pollutants in the processes. These approaches can be considered for a next work.

**Author Contributions:** Conceptualization, M.J.Á.-G., A.M.-G., and F.A.-G.; Investigation, M.J.Á.-G., A.M.-G., and J.R.L.-R.; Methodology, M.J.Á.-G. and F.A.-G.; Writing—original draft, M.J.Á.-G., A.M.-G., and J.R.L.-R.; Writing—review and editing, M.J.Á.-G., A.M.-G., and F.A.-G. All authors have read and agreed to the published version of the manuscript.

**Funding:** This research received no external funding.

**Conflicts of Interest:** The authors declare no conflict of interest.

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
