# Peer review of "Eco-Holonic 4.0 Circular Business Model to Conceptualize Sustainable Value Chain towards Digital Transition"

_sustainability, doi:10.3390/su12051889_

Round 1

Reviewer 1 Report

It was a great pleasure to read this paper. It is very well documented, with a high quality of presentation, on a hot topic, with impact on economy. It also takes in consideration the human footprint on the environment. The technical elements are very well presented.

I like the perspective on Real and virtual Eco-Holon and Eco-holarchy.

Digital transformation of value chain are the bases of the blockchain technology in the field of Eco-Holonic 4.0 circular business.

The multilevel and multiscale holon knowledge model is a revolutionary one.

Author Response

Dear reviewer,

Firstly, the authors would like to thank the reviewer for their work in reviewing the manuscript and for their suggestions on how to improve the paper. We appreciated the time that you spent in doing this. In the attached document you will find an answer to your comment.

Yours faithfully, the autors.

Reviewer 2 Report

This is a massive piece of work. Congratulations for that. The figures create a very good tool to help understanding the concepts and logic of the work. However there is a limit of how much information you can include into one figure. Now the figures 5 and 10 exceed the limit. It is technically impossible to follow all the details the you have included. Please, compile these figures from the major level components, and present details separately. Some of the details can also be presented in text form only. Concerning figure 6 it is difficult to understand where the components of the sustainability principles come from? What do you mean with the sentence ¨metrics of sustainability have been selected which are naturally inspired¨? Also, try to avoid repeating some lists of components in the text and figures,such as big data, IoT. cloud computing... At the end you might add a sentence about the prequisities of implementing of this approach. And who should take the initiative first? 

Author Response

Dear reviewer,

Firstly, the authors would like to thank the reviewer for their work in reviewing the manuscript and for their suggestions on how to improve the paper. We appreciated the time that you spent in doing this. The attached manuscript version includes modifications in highlighted of their recommendations.

In the following attached document is answered all comments referencing the text changes.

Yours faithfully, the autors
